# Secure Inference for Diffusion Models via Unconditional Scores

**Jaeyun Song**[1]**, Geondo Park**[2]**, Uigyu Kim**[2]**, Joonhyung Park**[2] **, Eunho Yang**[2,3]
KRAFTON[1], Korea Advanced Institute of Science and Technology (KAIST)[2], AITRICS[3]
`jaeyun.song@krafton.com, eunhoy@kaist.ac.kr`

## Abstract

As diffusion model-based services expand across various domains, safeguarding client data privacy has become increasingly critical. While fully homomorphic encryption and secure multi-party computation enable privacy-preserving inference, their high computational overhead poses challenges for large-scale diffusion applications. Recent work alleviates computational costs by substituting non-linear operations with low-degree polynomial approximations. While such relaxations reduce latency, they incur significant degradation in generative fidelity, and inference remains considerably slower than plaintext execution. To further accelerate secure inference while preserving performance, we explore more relaxed approximations and propose a score-correction framework that rectifies the conditional score shift induced by the relaxed approximation, rather than decreasing the approximation error itself. The key insight is that unconditional generation can be executed without approximation and thus provides a high-fidelity score signal. Leveraging this unconditional score as corrective guidance enables more relaxed approximations while preserving semantic and perceptual quality. In experiments, we demonstrate that our method significantly alleviates the performance degradation caused by relaxed approximations across various benchmarks.

## 1 Introduction

Modern conditional diffusion models show impressive performance in generating images and are employed across a wide range of applications. Notably, Stable Diffusion (Rombach et al., 2022), one of the leading models, can produce high-quality images from user prompts or visual inputs such as sketches and key points. Building on the recent success of diffusion models, there is an increasing interest in their application in privacy-sensitive fields, such as medical domain and content creation. In the medical field, they are utilized for translating MRI to CT, denoising medical images, and medical anomaly detections. Additionally, DALL·E (Ramesh et al., 2022) and Midjourney have become popular as commercial tools for content generation and creative industries.

Despite these advancements, the substantial computational costs of image generation typically necessitate processing on powerful remote servers rather than local devices. This reliance on external servers introduces potential privacy concerns, as both user inputs and corresponding outputs may be exposed to the model provider. These concerns are especially critical in the medical domain, where inputs consist of sensitive information such as patient profiles and medical images, and outputs also include critical data such as CT scans in image translation tasks or enhanced images in denoising tasks. While sending data to a remote server raises privacy concerns for users, performing inference on client devices, on the other hand, exposes the model provider to the risk of model leakage. This dual privacy challenge complicates the deployment of diffusion models in privacy-sensitive fields.

Faced with the issue, CipherDM (Zhao et al., 2024) is the first to incorporate secure multi-party computation (MPC) (Yao, 1986) into diffusion models. Since the computation of non-linear operations is a major efficiency bottleneck in MPC, CipherDM approximates the non-linear activations and exponential function with polynomials. Although this strategy achieves substantial latency reductions, it causes significant performance degradation, and the overall latency remains significantly higher than that of plaintext execution, indicating that further acceleration is still required.

To further decrease the latency while maintaining performance, we investigate more aggressive approximations compared to the previous works, and propose a method that corrects approximation-induced score corruption rather than merely reducing individual approximation errors. The key insight is that unconditional generation can be executed without approximation, providing a high-fidelity score distribution that can be used as corrective guidance. By leveraging this unconditional score to steer the corrupted conditional score toward its true distribution, our framework mitigates score displacement and enables more aggressive polynomial approximations while preserving generation quality. Empirical results show that our approach significantly improves performance on binary MNIST with CipherDM, and on Flickr8k (Hodosh et al., 2013) and MSCOCO (Lin et al., 2014) with Stable Diffusion (Rombach et al., 2022) under highly relaxed approximations, using MPC frameworks.

Our contribution is threefold:

- We investigate relaxed polynomial approximations for non-linear activations and the exponential function in diffusion models and observe operation-level latency reductions of 14–36% in a 2PC setting.
- We propose a novel score-guidance method that exploits the unconditional generation score to correct the approximation-corrupted conditional score.
- Through extensive experiments, we show that the proposed method significantly mitigates performance degradation caused by relaxed approximations across multiple datasets.

## 2 PRELIMINARIES

### 2.1 DIFFUSION MODEL

Diffusion models (Sohl-Dickstein et al., 2015) learn data distribution $p_{\text{data}}$ by denoising a variable following a normal distribution. Diffusion models consist of two main processes: a forward process, where the data distribution is progressively transformed into a Gaussian distribution by incrementally adding noise, and a reverse process, where the Gaussian distribution is gradually transformed back into the data distribution by iteratively removing noise, guided by signals predicted by a model. Therefore, the reverse process serves as the main mechanism for generating data. To learn the reverse process, the models are trained to produce the variables from noise over $T$ steps. Specifically, let $\mathbf{x}_T, \ldots, \mathbf{x}_1$ be variables where the initial noise, $\mathbf{x}_T$, follows Gaussian distribution and $\mathbf{x}_0 \sim p_{\text{data}}$. For the model parameter $\theta$, the previous variable is obtained by the reverse process as follows:

$$p_\theta(\mathbf{x}_{t-1}|\mathbf{x}_t) := \mathcal{N}\left(\mathbf{x}_{t-1}; \mu_\theta\left(\mathbf{x}_t, t\right), \mathbf{\Sigma}_\theta\left(\mathbf{x}_t, t\right)\right). \tag{1}$$

In DDPM (Ho et al., 2020), the mean and covariance are predicted as:

$$\mu_\theta\left(\mathbf{x}_t, t\right) = \frac{1}{\sqrt{\alpha_t}}\left(\mathbf{x_t} - \frac{\beta_t}{\sqrt{1-\bar{\alpha}_t}}\epsilon_\theta(\mathbf{x}_t, t)\right), \mathbf{\Sigma}_\theta\left(\mathbf{x}_t, t\right) = \beta_t\mathbf{I}, \tag{2}$$

where $\beta_1, \ldots, \beta_T$ are variance schedules, $\alpha_t = 1 - \beta_t$, $\bar{\alpha}_t = \prod_{s=1}^{t}\alpha_s$, and $\sigma_t = \sqrt{\beta_t}$. Since obtaining the mean via $\epsilon$ prediction ($\epsilon_\theta(\mathbf{x}_t, t)$) in Equation 2 is more effective compared to directly predicting the mean ($\mu_\theta\left(\mathbf{x}_t, t\right)$), models are trained to predict $\epsilon$.

However, diffusion models are computationally intensive, as the diffusion process operates at the pixel level, resulting in substantial memory and computational demands, particularly for high-resolution images. To reduce the computational cost, Latent Diffusion Model (LDM) (Rombach et al., 2022) performs diffusion processes in the latent space. This latent space has a lower dimensionality and is constructed using an autoencoder. During inference, the initial noise $\mathbf{z}_T$ in the latent space progressively denoised following the reverse process to obtain $\mathbf{z}_0$ which aligns with the latent space distribution induced by the data distribution and the autoencoder's encoder. Finally, $\mathbf{z}_0$ is decoded back into the original data space as $\mathbf{x}_0$ using the autoencoder's decoder.

### 2.2 PRIVACY-PRESERVING INFERENCE

Fully homomorphic encryption (FHE) (Gentry, 2009) and secure multi-party computation (MPC) (Yao, 1986) are widely adopted for privacy-preserving inference. FHE allows operations to be

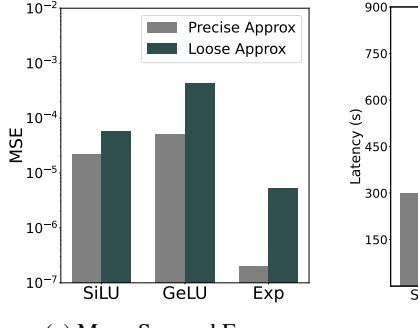 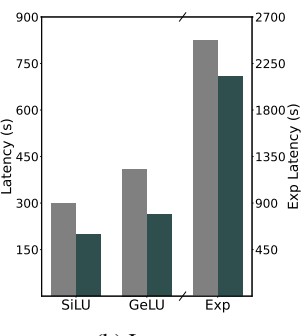 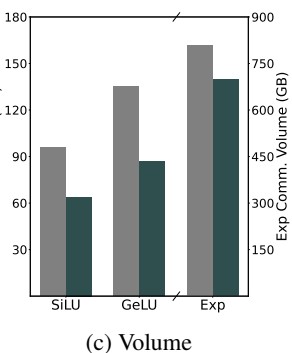

(a) Mean Squared Error      (b) Latency      (c) Volume

Figure 1: Comparison of mean squared error (MSE), latency (s), and communication volume (GB) for Stable Diffusion under two-party computation when reducing the polynomial degree or the number of piecewise segments for SiLU, GeLU, and the exponential function in softmax. Gray bars show precise approximations, and green bars show the relaxed degree/segment variants.

performed directly on encrypted data, ensuring that the decrypted result is nearly identical to the output of plaintext data. On the other hand, secure MPC allows multiple parties to jointly compute operations on combined data while preserving the privacy of each party's data. In secure MPC, secret data is divided into multiple shares and distributed among participants (Damgård et al., 2012), ensuring that reconstructing the original data is infeasible without all shares. Computations are executed on these shares, ensuring privacy is maintained throughout the process. Despite these guarantees, both approaches incur significantly higher latency than plaintext execution due to heavy computational overhead in FHE and intensive communication in MPC.

These private inference methods are commonly used in machine learning-as-a-service scenarios. As modern models now often exceed one billion parameters, latency has become a critical concern in such scenarios. Specifically, the evaluation of non-linear operations is a major efficiency bottleneck, motivating various approaches to mitigate the associated cost. Polynomial approximations and iterative algorithms have therefore been introduced to reduce computational overhead (Pang et al., 2024; Lu et al., 2025). However, for diffusion models, these methods often result in performance degradation, even on simple datasets (Zhao et al., 2024). The cumulative effect of small approximation errors can mislead the model, causing it to produce noisy and low-quality images. Therefore, adopting more relaxed polynomial approximations, such as further lowering the polynomial degree, while maintaining generation quality, remains a critical challenge for secure inference in diffusion models. To address this challenge, we propose a novel guidance strategy that leverages unconditional generation to correct the approximation-corrupted conditional score.

## 3 SECURE INFERENCE FOR DIFFUSION MODEL WITH SCORE GUIDANCE

We now investigate recent approximation techniques, explore a relaxed variant, and analyze how reducing the polynomial degree or the number of approximation segments affects both latency and performance in Section 3.1. Based on these observations, we introduce a method that corrects score shifts caused by approximation errors by leveraging the unconditional score in Section 3.2.

### 3.1 PERFORMANCE DEGRADATION UNDER RELAXED APPROXIMATIONS

First, we analyze the approximation accuracy and latency of non-linear operations in latent diffusion models, focusing on SiLU, GeLU (Hendrycks & Gimpel, 2016), and the exponential function. For evaluation, we report mean-squared error (MSE), total runtime, and communication volume for a single forward step of the latent diffusion model (excluding the text encoder and VAE decoder) of Stable Diffusion (Rombach et al., 2022) executed under a two-party computation (2PC) setting using CrypTen (Knott et al., 2021). We use Linux traffic control to simulate a local-area network with a one-way bandwidth of 3 Gbps and a 0.3 ms round-trip latency on an NVIDIA RTX A6000 GPU (48GB VRAM). For approximations, we adopt the polynomial schemes of BumbleBee (Lu et al., 2025) for SiLU, GeLU, and the exponential function, and those of CrypTen for other non-linear operations.

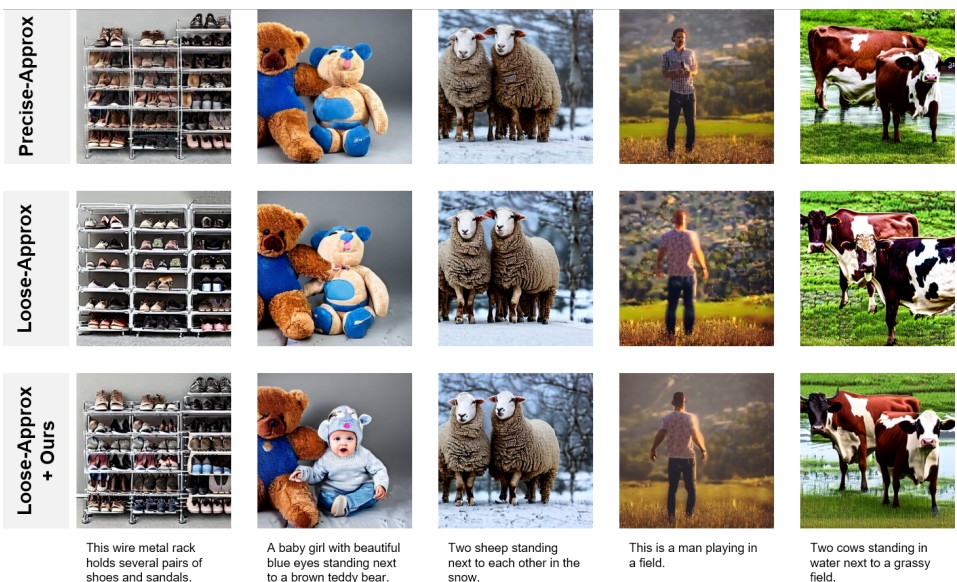

Figure 2: Comparison of text-to-image samples generated by a precisely approximated model (Precise-Approx), a loosely approximated model (Loose-Approx), and a loosely approximated model with our method (Loose-Approx + Ours).

For SiLU, in particular, the input domain is partitioned into four segments and approximated by low-degree polynomials: the first segment $[-8, -4)$ uses a quadratic (degree-2) polynomial, the second segment $[-4, 4)$ employs a degree-6 polynomial, and the remaining regions are clipped to constant or linear forms. The piecewise approximation is defined as

$$\text{Precise-SiLU}(x) = \begin{cases} 0, & \text{if } x < -8 \\ a_2 x^2 + a_1 x^1 + a_0, & \text{if } -8 \leq x < -4 \\ b_6 x^6 + b_4 x^4 + b_2 x^2 + b_1 x + b_0, & \text{if } -4 \leq x < 4 \\ x, & \text{if } x \geq 4, \end{cases}$$

where $a_0, a_1, a_2, b_0, \ldots, b_6$ denote coefficients optimized to minimize MSE. The corresponding approximations for GeLU and the exponential function are detailed in Appendix A.

To further reduce latency, lowering the polynomial degree or the number of piecewise segments are commonly used. Specifically, we reduce the number of segments for SiLU and GeLU from four to three, lower the polynomial degree of GeLU from six to four, and apply a fourth-degree polynomial approximation to the exponential function in the second segment. The relaxed piecewise approximation for SiLU is given by

$$\text{Loose-SiLU}(x) = \begin{cases} 0, & \text{if } x < -5.4 \\ c_8 x^8 + c_6 x^6 + c_4 x^4 + c_2 x^2 + c_1 x + c_0, & \text{if } -5.4 \leq x < 4 \\ x, & \text{if } x \geq 4, \end{cases}$$

where $c_0, \ldots, c_8$ are the optimal coefficients minimizing mean squared errors. The corresponding relaxed approximations for GeLU and the exponential function are provided in Appendix A.

As shown in Figure 1, these relaxed approximations reduce both runtime and communication cost, achieving 14.05–35.60% latency reduction and 13.51–33.33% communication reduction. However, this efficiency gain comes at the expense of accuracy: the approximation error increases by 2.55–26.25×, leading to significant fidelity degradation. As illustrated in Figure 2, naively applying relaxed approximations often produces blurred or structurally inconsistent images (for example, sheep with an excessive number of legs or misdrawn objects such as a stuffed bear when the prompt requests a baby girl). Additional qualitative comparisons are provided in Figure 5. One possible solution is to fine-tune the model after applying lower-degree approximations, but for large diffusion models, this approach is prohibitively expensive, requiring retraining models on hundreds of millions of images. To address this challenge, we aim to correct the score corruption induced by relaxed approximations at inference time, rather than merely reducing the approximation errors themselves.

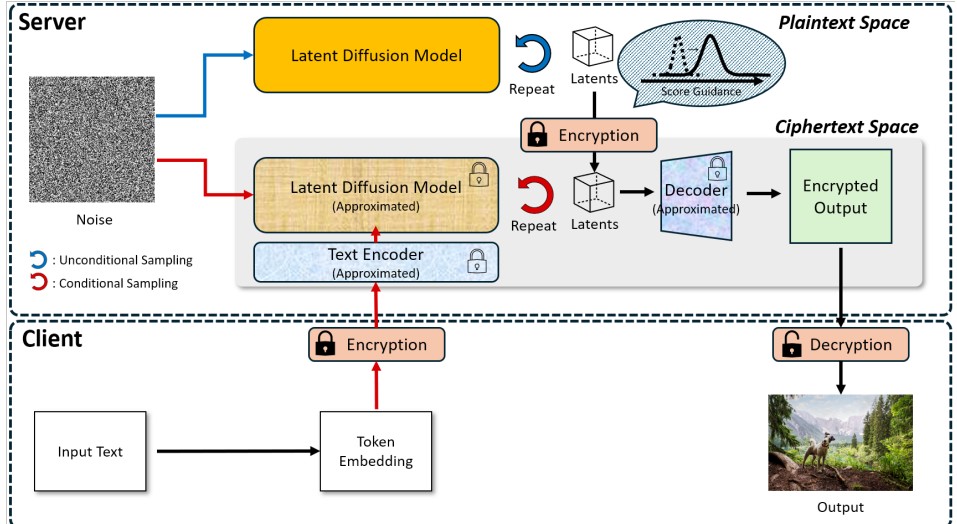

Figure 3: The overview of our scenarios. The client sends the input to the server securely (via secret sharing or homomorphic encryption), which processes and returns the result. Denoising process occurs securely, with the non-linear functions of the diffusion model approximated as linear. Next, our score guidance from unconditional scores is utilized to correct score shifts derived from approximation errors in conditional generation.

## 3.2 SCORE GUIDANCE FROM UNCONDITIONAL GENERATION

In the previous analysis, we observed that even small absolute approximation errors can accumulate across denoising steps, degrading image quality under more relaxed approximations. To mitigate this effect, we propose score guidance from unconditional generation, which refines the corrupted conditional scores by leveraging the score produced by unconditional generation. The key insight is that unconditional generation, executed entirely in the plaintext space, is free from approximation errors and can therefore provide a high-fidelity direction in the score space. The overall process is depicted in Figure 3.

Specifically, our goal is to adjust the conditional score to better align with the latent distribution obtained from generation without approximations. To derive the target distribution, we use an implicit discriminator that determines whether a given sample originates from generation without approximations:

$$\mathcal{D}^{\mathrm{im}}(\mathbf{z}_t) = \frac{p(o|\mathbf{z}_t)}{p(\hat{o}|\mathbf{z}_t)}, \tag{3}$$

where $o$ and $\hat{o}$ represent the labels for generation without approximations and that with approximations, respectively.

We then modify the latent to increase the likelihood of the discriminator identifying the sample as generated without approximations. The gradient of this objective is computed utilizing Bayes' rule:

$$\nabla_{\mathbf{z}_t}\mathcal{L}(\mathbf{z}_t) = \nabla_{\mathbf{z}_t}\left(-\log\mathcal{D}^{\mathrm{im}}(\mathbf{z}_t)\right) = -\nabla_{\mathbf{z}_t}\left(\log p(\mathbf{z}_t|o) - \log p(\mathbf{z}_t|\hat{o})\right).$$

Using Tweedie's formula (Robbins, 1992) and its score-based derivation (Luo, 2022), we approximate $-\nabla_{\mathbf{z}_t}\log p(\mathbf{z}_t|o) = \epsilon_\theta(\mathbf{z}_t, t)/\sigma_t$. To minimize $\mathcal{L}(\mathbf{z}_t)$, we update $\mathbf{z}_t$ in terms of $\epsilon$:

$$\begin{aligned}
\tilde{\epsilon}_\theta(\mathbf{z}_t, t) &= \epsilon_\theta(\mathbf{z}_t, t) + \gamma_t\sigma_t\nabla_{\mathbf{z}_t}\mathcal{L}(\mathbf{z}_t) \\
&= \epsilon_\theta(\mathbf{z}_t, t) - \gamma_t\sigma_t\nabla_{\mathbf{z}_t}\left(\log p(\mathbf{z}_t|o) - \log p(\mathbf{z}_t|\hat{o})\right) \\
&= \epsilon_\theta(\mathbf{z}_t, t) + \gamma_t\left(\epsilon_\theta(\mathbf{z}_t, t) - \hat{\epsilon}_\theta(\mathbf{z}_t, t)\right)
\end{aligned} \tag{4}$$

where $\gamma_t$ is the learning rate. Here, $\epsilon_\theta(\mathbf{z}_t, t)$, and $\hat{\epsilon}_\theta(\mathbf{z}_t, t)$ are $\epsilon$ predictions performed without approximations and with approximations, respectively.

---

**Algorithm 1** Score guidance from unconditional generation

---
1: **Input:** weight function $w(t)$, learning rate $\eta$, condition $y$, number of total timestep $T$, timestep offset $t_o$, classifier-free guidance strength $w_{\text{cfg}}$
2: $\mathbf{z}_T \sim \mathcal{N}(\mathbf{0}, \mathbf{I})$
3: $\mathbf{z}_T^{\text{unc}} \leftarrow \mathbf{z}_T$
4: $\mathbf{n} \sim \mathcal{N}(\mathbf{0}, \mathbf{I})$ if $t > t_o$, else $\mathbf{n} = \mathbf{0}$
5: **for** $t = T - t_o, T - 2t_o, \ldots, t_o, 0$ **do**
6: $\quad \epsilon_\theta^{\text{unc}} \leftarrow \epsilon_\theta(\mathbf{z}_{t+t_o}^{\text{unc}}, t + t_o)$
7: $\quad \mathbf{z}_t^{\text{unc}} \leftarrow \frac{1}{\sqrt{\alpha_{t+t_o}}} \left( \mathbf{z}_{t+t_o}^{\text{unc}} - \frac{\beta_{t+t_o}}{\sqrt{1-\bar\alpha_{t+t_o}}} \epsilon_\theta^{\text{unc}} \right) + \sqrt{\beta_{t+t_o}} \mathbf{n}$  $\qquad \triangleright$ Perform unconditional generation
8: $\quad \hat\epsilon_\theta^{\text{cfg}} \leftarrow (1 + w_{\text{cfg}})\hat\epsilon_\theta(\mathbf{z}_{t+t_o}, y, t + t_o) - w_{\text{cfg}}\hat\epsilon_\theta(\mathbf{z}_{t+t_o}, t + t_o)$  $\qquad \triangleright$ Apply classifier-free guidance
9: $\quad \tilde\epsilon_\theta \leftarrow \hat\epsilon_\theta^{\text{cfg}} + \eta \cdot w_{t+t_o}(\epsilon_\theta^{\text{unc}} - \hat\epsilon_\theta(\mathbf{z}_{t+t_o}, y, t + t_o))$  $\qquad \triangleright$ Conduct our score guidance
10: $\quad \mathbf{z}_t \leftarrow \frac{1}{\sqrt{\alpha_{t+t_o}}} \left( \mathbf{z}_{t+t_o} - \frac{\beta_{t+t_o}}{\sqrt{1-\bar\alpha_{t+t_o}}} \tilde\epsilon_\theta \right) + \sqrt{\beta_{t+t_o}} \mathbf{n}$
11: $\quad \mathbf{n} \sim \mathcal{N}(\mathbf{0}, \mathbf{I})$ if $t > t_o$, else $\mathbf{n} = \mathbf{0}$
12: **end for**
13: **Output:** $\hat{\mathbf{z}}_0$

---

For conditional generation, we combine classifier-free guidance with Equation 4 and $\tilde\epsilon_\theta(\mathbf{z}_t, y, t)$ is obtained as:

$$\tilde\epsilon_\theta(\mathbf{z}_t, y, t) = \hat\epsilon_\theta^{\text{cfg}}(\mathbf{z}_t, y, t) + \gamma_t \left( \epsilon_\theta(\mathbf{z}_t, y, t) - \hat\epsilon_\theta(\mathbf{z}_t, y, t) \right), \tag{5}$$

where $\hat\epsilon_\theta^{\text{cfg}}(\mathbf{z}_t, y, t)$ is computed using classifier-free guidance (Ho & Salimans, 2021).

However, since $y$ is encrypted, directly computing $\epsilon_\theta(\mathbf{z}_t, y, t)$ without approximations in the ciphertext space is infeasible. To address this, we use the unconditional score, $\epsilon_\theta^{\text{unc}}(\mathbf{z}_t^{\text{unc}}, t)$, as score guidance, where $\mathbf{z}_t^{\text{unc}}$ is acquired from unconditional generation without approximations:

$$\tilde\epsilon_\theta(\mathbf{z}_t, \mathbf{z}_t^{\text{unc}}, y, t) = \hat\epsilon_\theta^{\text{cfg}}(\mathbf{z}_t, y, t) + \gamma_t \left( \epsilon_\theta^{\text{unc}}(\mathbf{z}_t^{\text{unc}}, t) - \hat\epsilon_\theta(\mathbf{z}_t, y, t) \right). \tag{6}$$

An additional advantage of this design is that unconditional generation can be executed rapidly in the plaintext and precomputed offline, allowing the guidance signal to be reused during secure inference with negligible computational overhead.

A remaining challenge is that using the unconditional score may weaken prompt alignment. Since our goal is to generate high-quality images faithful to the conditioning prompt, the guidance of the unconditional score can become detrimental as the denoising process progresses toward $t = 0$, where the signal-to-noise ratio is highest. To mitigate this effect, we gradually reduce the guidance strength by decreasing the learning rate $\gamma_t$ as the timestep approaches $t = 0$. In practice, we adopt $\gamma_t = \eta w_t$, where $w_t = \sigma_t/\bar\alpha_t$. Note that we use $\eta = 0.1$ for all experiments.

With our score guidance, we observe that image quality is well preserved even when relaxed approximations are applied. As shown in Figure 2, compared to using relaxed approximations alone, our method generates sharper and more structurally consistent images, producing a feasible number of legs for the sheep and accurately incorporating the baby girl prompt in the second column. Additional qualitative comparisons are provided in Figure 5. These results indicate that our framework effectively mitigates the score corruption induced by relaxed approximations.

## 4 EXPERIMENTS

In this section, we evaluate our approach on the text-to-image benchmark datasets in a secure multi-party computation (MPC) setting. We first briefly describe the experimental setup and evaluation protocol, then compare our method with Stable Diffusion (Rombach et al., 2022) in Section 4.1 and with CipherDM (Zhao et al., 2024) in Section 4.2. Finally, we provide an ablation study about the learning rate in Section 4.3. To further validate the effectiveness of our approach, we compare our method against recent test-time refinement methods in Section 4.4 and evaluate its performance in the few-step sampling regime using consistency models in Section 4.5.

Table 1: Performance and latency comparison on MSCOCO. FID measures image quality, and CLIP-Score quantifies alignment between generated images and text prompts. Latency (s) is averaged over five runs in a 2PC setting with null prompts, and communication volume (GB) denotes the total communication cost. Latency and communication are measured for a single denoising step, while FID and CLIP-Score are computed using approximated models in plaintext due to resource constraints.

| Method | FID ($\downarrow$) | CLIP-Score ($\uparrow$) | Latency ($\downarrow$) | Comm. Volume ($\downarrow$) |
|---|---|---|---|---|
| Vanilla | 13.25 | 0.3254 | 0.15 | - |
| Precise-Approx | 13.28 | 0.3264 | 3696.45 | 1122.51 |
| Loose-Approx | 15.52 | 0.3240 | 3103.32 | 933.22 |
| Precise-Approx + **Ours** | **12.45** | **0.3274** | 3698.13 | 1122.51 |
| Loose-Approx + **Ours** | 14.20 | 0.3259 | 3105.38 | 933.22 |

Table 2: Quality comparisons on Flickr8k.

| Method | FID ($\downarrow$) | CLIP-Score ($\uparrow$) |
|---|---|---|
| Vanilla | 21.17 | 0.3369 |
| Precise-Approx | 20.85 | 0.3374 |
| Loose-Approx | 23.25 | 0.3347 |
| Precise-Approx + **Ours** | **19.63** | **0.3391** |
| Loose-Approx + **Ours** | 21.47 | 0.3372 |

Table 3: Ablation study about learning rate ($\eta$).

| Method | FID ($\downarrow$) | CLIP-Score ($\uparrow$) |
|---|---|---|
| Loose-Approx | 23.25 | 0.3347 |
| Loose-Approx + **Ours** ($\eta$=0.05) | 21.85 | 0.3367 |
| Loose-Approx + **Ours** ($\eta$=0.10) | 21.47 | 0.3372 |
| Loose-Approx + **Ours** ($\eta$=0.20) | **21.25** | **0.3376** |

**Experimental setting**  For image generation models, we use CipherDM (Zhao et al., 2024) and pre-trained Stable Diffusion v1.5 (Rombach et al., 2022) without any further fine-tuning. For CipherDM, we use ReLU activations and a DDIM sampler (Song et al.) with 50 denoising steps, running under SPU (Ma et al., 2023) with default configurations. For Stable Diffusion, the DDIM sampler (Song et al.) is utilized with a classifier-free guidance scale of 7.5 (Ho & Salimans, 2021) and 50 denoising steps. The latency measurement is conducted using CrypTen (Knott et al., 2021) in a 2PC configuration. For stable approximations, we additionally utilize widely used approximation techniques, detailed in Appendix A. We evaluate performance on binary MNIST for CipherDM, and MSCOCO (Lin et al., 2014) and Flickr8k (Hodosh et al., 2013) for Stable Diffusion.

**Evaluation protocol**  We follow the evaluation protocols of Zhao et al. (2024) for CipherDM and Kang et al. (2023) for Stable Diffusion. Image quality is assessed using Frechet Inception Distance (FID) (Heusel et al., 2017), while image alignment is measured using the F1-Score for CipherDM and the CLIP-Score (Radford et al., 2021) for Stable Diffusion. We generate 10,000 images for binary MNIST (5,000 per class), 30,000 for MSCOCO, and 8,000 for Flickr8k. To simulate a local-area network, we employ Linux traffic control with a one-way bandwidth of 3 Gbps and a 0.3 ms round-trip latency, and report the wall-clock time averaged over five repetitions. For Stable Diffusion, whose full inference latency is significantly high, we measure the latency of a single latent-diffusion step (excluding the text encoder and VAE decoder) using CrypTen, and generate images for evaluation in plaintext with models where non-linear operations are approximated. When computing latency and communication costs, the maximum operation for softmax stabilization is also calculated in plaintext due to its high computational overhead. For CipherDM, image generation is performed entirely under an MPC setting using SPU (Ma et al., 2023). Detailed experimental settings and evaluation protocols are provided in Appendix E.

**Model notation**  'Vanilla' refers to the original diffusion model. 'CipherDM' represents Zhao et al. (2024) with ReLU activations. 'Precise-Approx' indicates the diffusion model using precisely approximated non-linear operations, whereas 'Loose-Approx' denotes the model employing loosely approximated operations as described in Appendix A. '+Ours' represents the corresponding model combined with our proposed score guidance from unconditional generation.

Table 4: Performance and latency comparison on binary MNIST using CipherDM (ReLU) under SPU. FID and F1-Score are used to evaluate image quality and the alignment between generated images and conditions (0 or 1), respectively. Latency (s) and communication volume (GB) are averaged over 5 repetitions.

| Method | FID ($\downarrow$) | F1-Score ($\uparrow$) | Latency ($\downarrow$) | Comm. Volume ($\downarrow$) |
|---|---|---|---|---|
| Vanilla | 117.13 | 0.8575 | 14.12 | - |
| CipherDM | 316.56 | 0.0820 | 430.59 | 9.12 |
| CipherDM + **Ours** | **242.65** | **0.6000** | 442.23 | 9.14 |

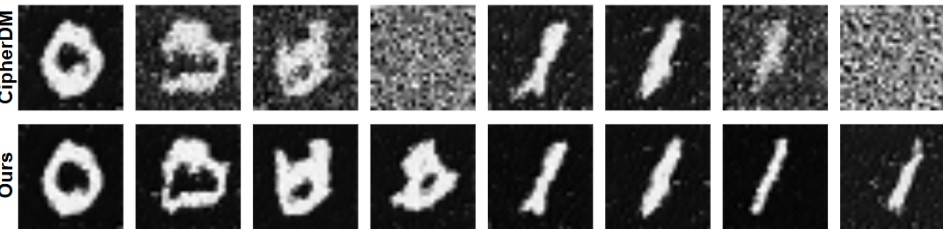

Figure 4: Comparison of samples between CipherDM and our method. The first and second rows show samples generated by 'CipherDM' and 'CipherDM+Ours', respectively.

## 4.1 EVALUATION ON STABLE DIFFUSION

We now evaluate our method with Stable Diffusion. In Table 1 and 2, relaxing the approximations accelerates inference, reducing latency by 16.04% and communication cost by 16.86%, but leads to substantial degradation in both image quality and prompt alignment. Remarkably, when our method is combined, the performance loss is significantly alleviated: for example, on MSCOCO the gap in FID between the precise approximation and the loosely approximated model decreases from 2.24 to 0.98, corresponding to a 56.25% reduction in quality degradation (similar improvements of up to 74.17% are observed on Flickr8k). Even with precisely approximated models, our method achieves performance that is comparable to or slightly better than the vanilla model. We speculate that this improvement arises since the vanilla model's score can deviate slightly due to random noise. These result implies that our score guidance approach effectively remedies the score corruption. For latency and communication costs, the overhead introduced by our method is negligible. The unconditional score is computed in plaintext, and the guidance computation requires only a single subtraction, addition, and scalar multiplication per denoising step. As a result, the overall latency and communication costs remain almost unchanged. A detailed breakdown of the latency and communication costs for each component is provided in Appendix D.

## 4.2 EVALUATION ON CIPHERDM

We also evaluate our method on binary MNIST using CipherDM. For unconditional generation, we use the average of class embeddings for labels 0 and 1. As shown in Table 4, our method improves the FID by over 70 compared to the original CipherDM and significantly boosts alignment from 0.0820 to 0.6000. In Figure 4, the samples generated by our method appear much clearer and less noisy than those from CipherDM. Even if the images generated by CipherDM are almost entirely noise, our method can recover clean and coherent digits. These results suggest that our method can effectively improve performance, even when the underlying approximation errors are large.

## 4.3 ABLATION STUDY FOR LEARNING RATE

We evaluate the sensitivity of the learning rate ($\eta$) on MSCOCO. In Table 3, as the learning rate increases, image quality is improved while maintaining the CLIP-Score. A potential concern is that using unconditional scores for guidance might degrade text alignment. However, since the guidance

Table 5: Comparisions with test-time methods.

| Method | FID ($\downarrow$) | CLIP-Score ($\uparrow$) |
|---|---|---|
| Loose-Approx | 23.25 | 0.3347 |
| Loose-Approx + PAG | 36.72 | 0.3171 |
| Loose-Approx + FKS | 24.42 | 0.3359 |
| Loose-Approx + **Ours** | **21.47** | **0.3372** |

Table 6: Performance with PCM.

| Method | FID ($\downarrow$) | CLIP-Score ($\uparrow$) |
|---|---|---|
| Loose-Approx (50 steps) | 23.25 | 0.3347 |
| Loose-Approx + PCM (8 steps) | 36.69 | 0.3264 |
| Loose-Approx + PCM (4 steps) | 53.80 | 0.3183 |
| Loose-Approx + PCM + **Ours** (8 steps) | 27.23 | 0.3350 |
| Loose-Approx + PCM + **Ours** (4 steps) | 36.69 | 0.3264 |

strength is designed to decrease as the denoising signal grows, text alignment remains unaffected. These results demonstrate that our approach effectively corrects score shifts without compromising text alignment across a broad range of learning rates.

### 4.4 COMPARISON WITH TEST-TIME APPROACHES

To further validate the effectiveness of our approach, we compare it against several test-time methods, including degraded-sample–based refinement (Ahn et al., 2024), and a test-time scaling strategy (Singhal et al., 2025). Under the loose approximation setting, all alternatives perform comparably to or worse than the loose approximation alone (in Table 5), since these methods rely on approximated conditional models. Specifically, PAG (Ahn et al., 2024) requires generating mildly degraded samples, but under approximation errors, the degraded samples become severely corrupted, leading to performance degradation. FKS (Singhal et al., 2025), despite using multiple particles, also depends on an approximated verifier model. Thus, verification becomes unreliable and limits its effectiveness. In contrast, our method utilizes unconditional generation and therefore avoids approximation errors.

### 4.5 EVALUATION WITH CONSISTENCY MODELS

We evaluate our method in the few-shot regime using Phased Consistency Model (PCM) (Wang et al., 2024) with 4 and 8 sampling steps. In Table 6, vanilla PCM suffers from significant performance degradation at lower steps due to approximations. However, integrating our method yields significant improvements, specifically gaining approximately 17 FID at 4 steps and 9 FID at 8 steps. These results indicate that our approach provides a practical solution for achieving both efficiency and accuracy in secure inference.

## 5 RELATED WORK

### 5.1 CONDITIONAL DIFFUSION MODEL

With the development of diffusion-based models (Sohl-Dickstein et al., 2015; Ho et al., 2020; Song et al.; Nichol & Dhariwal, 2021), significant progress has been made in various conditional image generation tasks, such as text-conditioned image synthesis (Saharia et al., 2022) and masked-region image editing (Lugmayr et al., 2022). Various open source models and commercial models such as DALLE (Ramesh et al., 2022), Imagen (Saharia et al., 2022), and GLIDE (Nichol et al., 2022) have contributed to this development. Recently, classifier-free guidance (Ho & Salimans, 2021) simplifies the conditional image generation process by directly incorporating class conditions into the diffusion model, eliminating the need for an external classifier. Based on this work, Stable Diffusion (Rombach et al., 2022) has excelled in generating high-quality images efficiently, leading to the exploration of various applications. For test-time scaling, FKSteering (Singhal et al., 2025) produces samples that are more aligned with desired preferences by simultaneously maintaining multiple particles and iteratively refining them using verifiers. However, in secure inference settings, maintaining multiple particles can be computationally expensive, and verification may be unreliable because the verifiers themselves must also be approximated. For sampling guidance, prior approaches (Hong et al., 2023; Ahn et al., 2024) have improved the quality of generated images by refining the scores during inference. These methods encourage models to synthesize more desirable samples by generating undesirable ones and guiding the target scores away from those of undesirable samples. While our approach shares similarities with these methods in promoting the generation of desirable samples, it introduces several key differences. First, our focus is on generating the samples that correspond

to images produced without approximations, since generating target samples inherently involves approximation errors.

## 5.2 PRIVATE INFERENCE

Fully homomorphic encryption (FHE) (Gentry, 2009; Cheon et al., 2017) and secure multi-party computation (MPC) (Yao, 1986; Evans et al., 2018; Damgård et al., 2019; Goldreich et al., 2019) are widely used in private inference. In contrast to differential privacy (DP) (Dockhorn et al., 2023; Jiang et al., 2025), which focuses on protecting training datasets, these cryptographic approaches aim to protect user inputs from model providers during the inference. CryptoNets (Gilad-Bachrach et al., 2016) pioneered the use homomorphic encryption for small neural networks in image classification using polynomial approximations. To exploit the complementary strengths of FHE and MPC, several hybrid frameworks (Juvekar et al., 2018; Reagen et al., 2021; Hao et al., 2022; Pang et al., 2024; Li et al., 2024; Lu et al., 2025) apply FHE to linear operations and MPC to non-linear operations. More recently, a non-interactive approach (Zhang et al., 2025) has been proposed to further reduce both the computational burden on clients and the associated communication costs. For efficient evaluation of non-linear functions, prior work (Li et al., 2023; Zeng et al., 2023; Zhang et al., 2023) commonly employs piecewise polynomial approximations for activations such as SiLU and GeLU (Hendrycks & Gimpel, 2016), or iterative algorithms such as Newton or Goldschmidt algorithms (Goldschmidt, 1964) for inverse and square-root computations. For diffusion models, HE-Diffusion (Chen & Yan, 2024) and CipherDM (Zhao et al., 2024) enable privacy-preserving image generation using FHE and MPC, respectively. However, HE-Diffusion leaves the text prompt unencrypted or unshared, leading to potential privacy risks, whereas CipherDM suffers significant performance degradation due to approximation errors. To address these limitations and further accelerate inference, we adopt more relaxed approximations and propose a method that corrects conditional scores by leveraging unconditional scores.

## 6 CONCLUSION

In this paper, we introduce relaxed polynomial approximations and propose a new framework to mitigate the performance degradation they induce. Specifically, we present a score-guidance approach that corrects conditional scores by leveraging unconditional generations computed without approximations. Experimental results show that, when combined with our approximations, diffusion models can substantially recover performance while incurring only negligible additional costs.

**Limitations** In this study, we enable diffusion models to generate high-quality images in private inference while substantially reducing latency. However, the absolute latency of a single forward pass remains too high for practical deployment in real-world applications, and the performance evaluation of large models is conducted in plaintext. By improving the performance comparable to the original one, we believe this research paves the way for practical private inference.

## ACKNOWLEDGMENTS

This work was partly supported by Institute for Information & communications Technology Technology Planning & Evaluation(IITP) grant funded by the Korea government(MSIT) (RS-2019-II190075, Artificial Intelligence Graduate School Support Program(KAIST), No.RS-2025-02305581, Development of Vision-Language Model (VLM)-Based Intelligent Video Security Monitoring Technology, No. 2022-0-00984, Development of Artificial Intelligence Technology for Personalized Plug-and-Play Explanation and Verification of Explanation).

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

# A    APPROXIMATION DETAILS

For the polynomial approximations of non-linear functions such as SiLU, GeLU (Hendrycks & Gimpel, 2016), and the exponential function in softmax, we determine the coefficients by minimizing the mean squared error (MSE) over a targeted input interval. To select this interval, we perform a discrete search to identify the boundary that minimizes the MSE within the original segment. For example, in SiLU, the lower bound of $-5.4$ is obtained by treating the boundary as a hyperparameter and searching over $\{-7.9, -7.8, \ldots, 3.9\}$, minimizing the MSE in the range $[`8, 4)$. For numerical stability during secure inference, we also apply stabilization techniques to GroupNorm (Wu & He, 2018) and softmax.

**SiLU**    In BumbleBee (Lu et al., 2025), SiLU is partitioned into four input segments and approximated by low-degree polynomials as given by

$$\text{Precise-SiLU}(x) = \begin{cases} 0, & \text{if } x < -8 \\ a_2 x^2 + a_1 x^1 + a_0, & \text{if } -8 \le x < -4 \\ b_6 x^6 + b_4 x^4 + b_2 x^2 + b_1 x + b_0, & \text{if } -4 \le x < 4 \\ x, & \text{if } x \ge 4, \end{cases}$$

where $a_2 = -0.0055465625580307, a_1 = `0.0819767021525476, a_0 = `0.3067541139982155,$ $b_6 = 0.0002743776353465, b_4 = `0.011113046708173, b_2 = 0.2281430841728270, b_1 = 0.5,$ $b_0 = 0.0085064025895951$. For the simplified version, SiLU is reduced to three input segments and expressed as

$$\text{Loose-SiLU}(x) = \begin{cases} 0, & \text{if } x < -5.4 \\ c_8 x^8 + c_6 x^6 + c_4 x^4 + c_2 x^2 + c_1 x + c_0, & \text{if } -5.4 \le x < 4 \\ x, & \text{if } x \ge 4, \end{cases}$$

where $c_8 = -0.00000528080234, c_6 = 0.000397939016, c_4 = -0.0118558684,$ $c_2 = 0.228920817, c_1 = 0.500011828, c_0 = 0.00882801754.$

**GeLU**    In BumbleBee, GeLU is partitioned into four input segments and approximated by low-degree polynomials as given by

$$\text{Precise-GeLU}(x) = \begin{cases} 0, & \text{if } x < -5 \\ a_3 x^3 + a_2 x^2 + a_1 x^1 + a_0, & \text{if } -5 \le x < -1.97 \\ b_6 x^6 + b_4 x^4 + b_2 x^2 + b_1 x + b_0, & \text{if } -1.97 \le x < 3 \\ x, & \text{if } x \ge 3, \end{cases}$$

where $a_3 = `0.0110341340306157, a_2 = `0.1180761295118195, a_1 = `0.4222658115198386,$ $a_0 = `0.5054031199708174, b_6 = 0.0018067462606141, b_4 = `0.037688200365904,$ $b_2 = 0.3603292692789629, b_1 = 0.5, b_0 = 0.0085263215410380$. For the simplified version, GeLU is partitioned into three input segments and approximated by

$$\text{Loose-GeLU}(x) = \begin{cases} 0, & \text{if } x < -2.1 \\ c_4 x^4 + c_2 x^2 + c_1 x + c_0, & \text{if } -2.1 \le x < 3 \\ x, & \text{if } x \ge 3, \end{cases}$$

where $c_4 = -0.01625818, c_2 = 0.30074739, c_1 = 0.49693733, c_0 = 0.03222096.$

**Exponential in softmax**    Similar to BumbleBee, the exponential function is partitioned into two input segments and approximated by

$$\text{Precise-Exp}(x) = \begin{cases} 0, & \text{if } x < -14 \\ \left(1 + \frac{x}{2^8}\right)^{2^8}, & \text{if } x \ge -14. \end{cases}$$

Since the input to the exponential function in the softmax is always non-positive, we further approximate the second segment with a fourth-degree polynomial,

$$\text{Loose-Exp}(x) = \begin{cases} 0, & \text{if } x < -4.8 \\ c_4 x^4 + c_3 x^3 + c_2 x^2 + c_1 x + c_0, & \text{if } -4.8 \le x \le 0, \end{cases}$$

where $c_4 = 0.00488676, c_3 = 0.06755453, c_2 = 0.36066246, c_1 = 0.91766503, c_0 = 0.98805643$. In CipherDM (Zhao et al., 2024), the exponential function is approximated with Chebyshev polynomial after mapping $x$ from $[-14, 0]$ to $[-1, 1]$ as:

$$\text{CipherDM-Exp}(x) = \begin{cases} 0, & \text{if } x < -14 \\ \sum_{i=0,\ldots,7} c_i x^i, & \text{if } -14 \leq x \leq 0, \end{cases}$$

where $c_7 = 0.6377824, c_6 = 0.66811776, c_5 = -0.41698096, c_4 = -0.27598784, c_3 = 0.28150564, c_2 = 0.09208474, c_1 = -0.0238906, c_0 = -0.01111495$.

**GroupNorm** For numerical stability, in GroupNorm and LayerNorm (Ba, 2016), we scale down the variance of the input tensor by a sufficiently large factor before normalization using the fact that $\frac{\mathbf{x}-\text{mean}(\mathbf{x})}{\text{deviation}(\mathbf{x})} = \frac{\mathbf{x}/d-\text{mean}(\mathbf{x}/d)}{\text{deviation}(\mathbf{x}/d)}$ for any non-zero $d$. To determine $d$, we increase $d$ as the number of channels in the group decreases, leveraging the fact that variance is proportional to the number of samples. Specifically, in the text encoder, we set $d = 8$ for LayerNorm. In the latent diffusion model, we use $d = \max(\sqrt{(64 \cdot 64 \cdot 30)/d_x}, 1)$ for GroupNorm, where $d_x$ denotes the dimension of the channels of each group, and $d = 1$ for LayerNorm. In the decoder of the compression model, we apply $d = \sqrt{(512 \cdot 512 \cdot 256)/d_x}$ for GroupNorm.

**Softmax** In softmax, we subtract the maximum input value before computing softmax, as $\frac{\exp(x_i)}{\sum_j \exp(x_j)} = \frac{\exp(x_i - x_{\max})}{\sum_j \exp(x_j - x_{\max})}$ for $x_{\max} = \max_j x_j$. Since the model processes 4096 visual tokens in Stable Diffusion (Rombach et al., 2022), we divide the input tensor by the square root of the number of tokens after applying the exponential but before applying the reciprocal. Thus, we compute the softmax as: $\frac{\exp(x_i - x_{\max})}{\sum_j \exp(x_j - x_{\max})} = \frac{\exp(x_i - x_{\max})/\sqrt{d}}{\sum_j (\exp(x_j - x_{\max}/\sqrt{d}))}$, where $d$ is the number of tokens.

# B EXPERIMENT UNDER WIDE AREA NETWORK

Using Linux traffic control, we simulate a Wide Area Network (WAN) with 400Mbps bandwidth and a 4ms ping time to evaluate our method on CipherDM. As shown in Table 7, when the unconditional score is computed during inference, the latency overhead introduced by our method is only 1.12%. Moreover, because the unconditional score can be precomputed offline, excluding this cost reduces the latency overhead to just 0.02%, which is negligible relative to the total runtime.

Table 7: Performance and latency comparison on binary MNIST using CipherDM (ReLU) under SPU. FID and F1-Score are used to evaluate image quality and the alignment between generated images and conditions (0 or 1), respectively. Latency (s) and communication volume (GB) are averaged over 5 repetitions under Wide Area Network (WAN).

| Method | Latency ($\downarrow$) | Comm. Volume ($\downarrow$) |
|---|---|---|
| Vanilla | 14.12 | - |
| CipherDM | 1250.87 | 9.11 |
| CipherDM + **Ours** | 1265.26 | 9.13 |

# C IMPACT ANALYSIS OF ENCODER AND DECODER APPROXIMATIONS

To investigate the individual contributions of text encoder and VAE decoder approximations to the overall performance, we conduct experiments by replacing them with exact computations while maintaining approximations in the latent diffusion model (LDM). As shown in Table 8, eliminating these approximations reduces the quality degradation by approximately 1.3 FID. Notably, under this configuration, our proposed method fully recovers the performance of the vanilla model. This result underscores that our score guidance effectively addresses the predominant approximation errors that arise within the latent diffusion component.

Table 8: Ablation study on the impact of exact text encoder and VAE decoder computations on MSCOCO. FID ($\downarrow$) and CLIP-Score ($\uparrow$) are reported for quality and alignment assessment.

| Method | FID ($\downarrow$) | CLIP-Score ($\uparrow$) |
|---|---|---|
| Vanilla | 13.25 | 0.3254 |
| Loose-Approx | 15.52 | 0.3240 |
| Loose-Approx + Exact-Enc-Dec | 14.22 | 0.3257 |
| Loose-Approx + Exact-Enc-Dec + **Ours** | **13.10** | **0.3269** |

## D    COMPUTATIONAL PROFILING

We analyze the computational overhead of each module to identify the primary bottleneck. Table 9 presents the latency and communication costs measured in a 2PC setting. While the text encoder's contribution is negligible and the VAE decoder shows moderate costs, the latent diffusion model (LDM) represents the majority of the total latency. Given that the LDM requires iterative evaluation, improving its approximation efficiency is critical for practical deployment. These observation justifies our strategy of applying loose approximations specifically to the LDM to achieve substantial speedups.

Table 9: Latency (s) and communication cost (GB) breakdown by component in a 2PC configuration. Measurements were conducted to evaluate the relative overhead of each module.

| Module | Latency ($\downarrow$) | Comm. Volume ($\downarrow$) |
|---|---|---|
| Text-Encoder | 40.34 | 7.76 |
| VAE-Decoder | 2668.66 | 763.40 |
| Latent Diffusion (1-step) | 3696.45 | 1122.51 |

## E    EXPERIMENTAL SETUP

We now provide more detailed experimental settings for the evaluation protocol and inference details.

**Evaluation protocol**    We follow the evaluation protocols of Zhao et al. (2024) for CipherDM and Kang et al. (2023) for Stable Diffusion. To compute the F1 score, we fine-tune a ResNet-18 (He et al., 2016) pre-trained on ImageNet (Deng et al., 2009) by replacing and training only the classifier head, while freezing all other parameters. The model is fine-tuned on the binary MNIST training set (digits 0 and 1) for 5 epochs using the Adam optimizer (Kingma, 2014) with a learning rate of 0.001. The classification accuracy of the fine-tuned ResNet-18 is nearly perfect on the test set, ensuring reliable F1 evaluation. To compute the FID score on binary MNIST, we use only the training set containing digits 0 and 1. For latency estimation in CipherDM, we use Intel(R) Xeon(R) Gold 6226R CPU and 480GB RAM for CipherDM. For Stable Diffusion, we utilize Intel(R) Xeon(R) Gold 6226R CPU, 256GB RAM, and NVIDIA RTX A6000 GPU (48GB VRAM).

**Inference details**    We adopt a learning rate of $\eta = 0.1$ for all experiments. Since the architecture of CipherDM uses class embeddings for conditioning, we use the mean of all class embeddings during unconditional generation.

## F    THREAT MODELS

We follow a standard threat model commonly adopted in private inference literature, assuming semi-honest adversaries and computationally bounded participants. In our implementation, we adopt two representative MPC frameworks: CrypTen (Knott et al., 2021), which operates in a 2-party setting, and SPU (Ma et al., 2023), which uses a 3-party setting. Our system design and empirical evaluations are aligned with the operational guarantees of these frameworks.

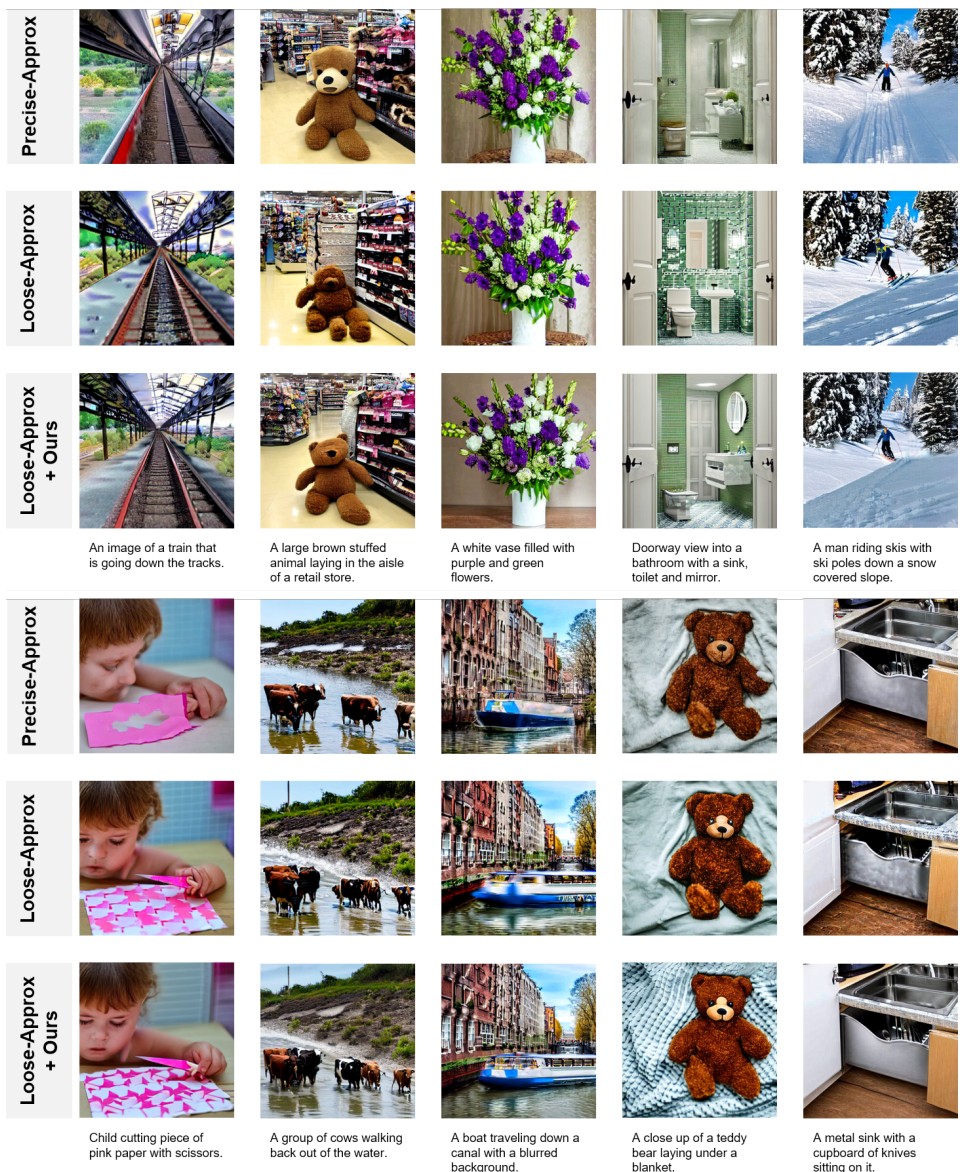

Figure 5: Comparison of text-to-image samples generated by a precisely approximated model (Precise-Approx), a loosely approximated model (Loose-Approx), and a loosely approximated model with our method (Loose-Approx + Ours).

## G  MORE EXAMPLES

Below, we provide visualizations of generation examples for extended qualitative assessment in Figure 5. When the approximation is relaxed, the model struggles to produce fine-grained components, such as fingers or the precise shapes of moving objects, and often yields blurred images compared with the precisely approximated model. In contrast, when combined with our method, the model successfully generates clear fingers, well-defined object contours, and sharp images, demonstrating that our approach effectively mitigates the score corruption introduced by relaxed approximations.

