# OpenReview forum: "Secure Inference for Diffusion Models via Unconditional Scores"
_ICLR.cc/2026/Conference — ICLR 2026 Poster_

### Official Review · Reviewer_tfMV · 2025-10-23

**Soundness:** 3
**Presentation:** 4
**Contribution:** 3
**Rating:** 6
**Confidence:** 3

**Summary:**

The paper studies efficient implementation of inference for diffusion models (IDM) under secure multi-party computation (MPC). A computational challenge encountered in previous work (CipherDM, Zhao et al., 2024) is that inference requires the computation of non-linear functions on the ciphertext, which is known to be computationally intensive. CipherDM addressed this issue by switching to polynomial approximations of the non-linear function, and so improving the scalability.

This paper revisits CipherDM with a coarser polynomial approximation, showing improved efficiency at the cost of a further deterioration in output quality. To mitigate the drop in output quality, the authors propose to augment the inference process by using unconditional and exact generation in plaintext to “steer” the diffusion process. The key idea is that the unconditional generation has no sensitive content, and so this signal can be computed fast and exactly in plaintext, and can consequently be used to guide the diffusion process to correct for errors induced by the polynomial approximation at a low computational overhead. Experiments confirm that the augmented signal can compensate for much of the approximation error, even to the point of beating the exact baseline in some cases.

**Strengths:**

1. The paper is well-written with a clear problem statement supported by nice figures.
2. The key idea of using unconditional sampling in plaintext space to (with small overhead) inform the generation process is natural and clever.
3. The problem is well-motivated and fits the scope of ICLR.
4. The experimental results are strong, and suggest that activation functions can be polynomially approximated at little to no cost in image quality (Table 1 and 2).
5. The empirical setup is given in enough detail (including an attached code artifact) to be reproducible.

**Weaknesses:**

1. The alternative of fine-tuning the DM with poly-approximated activations is not considered in depth. Since the main contribution of the paper is to make secure inference for DM more efficient, it would be fitting to have a more extensive discussion of this alternative approach and its accuracy. Line 212-214 correctly makes the point that the corresponding fine-tuning of large DMs is expensive, but one could nevertheless imagine large industry actors training DMs for the express purpose of enabling high-performance secure inference. If there is a gain in output accuracy from fine-tuning, they may find the associated cost worth taking on. See later questions.
2. (Conditioned on the validity of the previous point) If fine-tuning the DM with poly-approximated activations can achieve better accuracy than the proposed method in this paper, then the impact of the contribution is reduced. The authors improve the accuracy/efficiency trade-off for MPC-IDM only in the particular case where a diffusion model is not trained with MPC in mind, and is then “retrofitted” for the purpose. Nevertheless, I concede that (1) this might be an important case, and (2) in the case of precise approximation the empirical work suggests that fine-tuning the model may be excessive.
3. A minor point. While it is mentioned that CipherDM (Zhao et al., 2024) also employed polynomial approximation of activation functions (line 49) to improve performance for secure inference, the details of their approximation are (as far as I can tell) not discussed. Section 3.1 discusses polynomial approximations (“precise” and “loose” variants), but does not relate this to CipherDM. For completeness, it could be nice to include their approximation scheme in the appendix. It would also be useful from the point of view of having more interpretability of Table 4, here the signal steering, while helpful, appears to still lag far behind the baseline---is it due to the coarseness of the approximation in CipherDM? I don't think this can be inferred from the paper currently.

**Questions:**

I find the paper’s topic of efficient secure inference for diffusion models interesting, its proposed technique clever and its writing of high quality. The empirical work is clean and appears reproducible. My only concern is related to fine-tuning as an alternative to the “signal steering” approach proposed, which I think is dismissed with insufficient motivation. As things stand, I lean towards acceptance, and feel more inclined to raising my score than lowering it.

I have some questions. As they are a few, feel free to give brief answers where you see fit.
1. My understanding of your approach is that you use the signal from the “uncondtional+exact” latent diffusion model to steer the approximated latent diffusion model to correct for approximation errors. Nevertheless, from Figure 3 it seems like your text encoder and decoder are also approximated. Did you try investigating how much of the drop in image quality is due to these particular approximations for your approach? If your technique corrects for most of the approximation error in the latent model, then I would expect the performance to increase if the encoder/decoder are exact, at least in the case of the coarse approximation.
2. Related question: with no approximations, how much (roughly) of the latency is driven by the encoder/decoder step from Figure 3? My understanding is that Table 1 and 2 do not measure their influence, so I am curious how important they are to approximate.
3. What is the cost for instead fine-tuning the DM with polynomially-approximated activations driven by? Is it only the cost of fine-tuning a “normal” DM, or is there some additional cost that is imposed from the setting of secure inference?
4. You observe in Table 1 and 2 that your signal steering can not only compensate for the approximation error, but improve on the baseline if the approximation is precise enough. You hypothesize in Section 4.1 that this improvement could be driven by that your steering has a stabilizing influence on the process. Did you consider using your steering technique on the *exact* baseline itself? It would be interesting to see if the 'vanilla' model + steering also led to some improvement.
5. In the related work, you mention sampling guidance (Hong et al., 2023; Ahn et al., 2024). Could your idea of "signal steering" with the exact model in plaintext be combined with their techniques, or are they incompatible? Can "sampling guidance" be implemented without additional computation on ciphertext?
6. While it might be computationally expensive in general, did you consider testing fine-tuning of the DM for e.g., the MNIST dataset? E.g., fine-tune CipherDM and see how it compares with the results in Table 4?
7. Is the last sentence of the caption to Table 1 related to FID and CLIP-Score important for interpreting the accuracy results (computations being done in plaintext)? I would expect the reported image quality to be independent of this detail.

---

> ### Author Response · Authors · 2025-11-20
> **Response to Reviewer tfMV (1/2)**
>
> We sincerely appreciate your thoughtful comments, your positive evaluation of our work, and the time you invested in reviewing our manuscript. We are very grateful for your encouraging feedback. Below, we respond to each of your comments one by one as follows.
>
> ---
>
> ### **1) Details in CipherDM experiments**
>
> We agree that including the approximation scheme used in CipherDM would improve completeness, and we will add the details to the appendix in our revision. Specifically, CipherDM employs a Chebyshev polynomial approximation of degree 7 for the exponential function, using two segments (values below −14 are treated as zero).
>
> Regarding our experimental design, CipherDM already proposes its own polynomial approximation method, and there is a significant performance degradation observed under their setting. Thus, we did not apply our loose approximation. Instead, we directly used CipherDM’s approximation and incorporated our guidance on top of it. This results in a very slight latency increase due to the unconditional generation, but improves performance significantly.
>
> ---
>
> ### **2) Effect of approximations in text encoder and decoder**
>
> We evaluate the contribution of the text encoder and VAE decoder approximations by replacing them with exact computations while keeping the latent diffusion model approximated. As shown below, removing these approximations reduces the performance improvement by approximately 1.3 FID, and under this setting, **our method fully recovers the performance of the vanilla model**. This confirms that our method effectively corrects the dominant approximation errors originating from the latent diffusion model.
>
> | MSCOCO | FID | CLIP-Score |
> |---|---|---|
> | Vanilla | 13.25 | 0.3254 |
> | Loose-Approx | 15.52 | 0.3240 |
> | Loose-Approx w/ Exact-Enc-Dec | 14.22 | 0.3257 |
> | Loose-Approx w/ Exact-Enc-Dec +Ours | **13.10** | **0.3269** |
>
> To assess their computational significance, we measure the latency and communication cost of each component. The text encoder contributes negligibly to the total cost, while the VAE decoder is moderately expensive. However, **the latent diffusion model overwhelmingly dominates the latency, especially in a 50-step sampling regime**. Therefore, reducing approximation strength in the latent diffusion model is essential for practical efficiency. For this reason, we apply the loose approximation only to the latent diffusion component.
>
> | Method | Latency(s) | Comm.Vol.(GB) |
> |---|---|---|
> | Text-Encoder | 40.34 | 7.76 |
> | VAE-Decoder | 2668.66 | 763.40 |
> | LatentDiffusion (1-step) | 3696.45 | 1122.51 |
>
> ---
>
> ### **3) Combination with vanilla model**
>
> We thank the reviewer for the insightful suggestion. However, since our method is specifically designed to target and reduce the score shift induced by approximation errors, we expect its improvement on vanilla to be limited.
>
> ---
>
> ### **4) Combination with other test-time correction approaches**
>
> Our method targets the shift introduced by approximation errors, whereas PAG [1] guides the sampling trajectory toward regions of high density within the approximated model’s space. Since the two approaches address different aspects of the problem, **they are conceptually compatible**. However, PAG relies on generating “bad” but moderately degraded samples. Under approximation errors, the degradation becomes excessively severe, and using such samples results in further quality deterioration. In contrast, unconditional generation in our method is unaffected by this issue, allowing **our guidance to partially mitigate the degradation when combined with PAG**. Furthermore, when PAG is used together with CFG, an additional degraded sample must be generated in addition to the conditional and unconditional samples. This increases the communication cost by approximately 1.5×, resulting in a slower overall process. We note that, since using PAG alone severely collapses sample quality under approximation and the original PAG paper reports that PAG+CFG yields better performance than PAG alone, we adopt PAG+CFG as the baseline configuration for our experiment.
>
> | Flickr8k | FID | CLIP-Score |
> |---|---|---|
> | Vanilla | 21.17 | 0.3369 |
> | Loose-Approx | 23.25 | 0.3347 |
> | Loose-Approx +PAG(with CFG) [1] | 36.72 | 0.3171 |
> | Loose-Approx +PAG(with CFG) +Ours | 31.91 | 0.3196 |
>
> [1] Ahn, Donghoon, et al. "Self-rectifying diffusion sampling with perturbed-attention guidance." ECCV, 2024.

---

> ### Author Response · Authors · 2025-11-20
> **Response to Reviewer tfMV (2/2)**
>
> ### **5) The potential performance difference between plaintext and ciphertext**
>
> Except for approximation errors, the difference between plaintext and MPC execution can introduce additional numerical deviations. MPC relies on fixed-point arithmetic, which has a narrower representable range compared to floating-point, and ciphertext–ciphertext multiplications require a truncation step to rescale precision. Both factors can introduce small errors. However, when using 32-bit precision, fixed-point representation provides sufficient accuracy to avoid meaningful degradation. Moreover, recent truncation protocols discard only the lower insignificant bits, ensuring that the resulting numerical error remains minimal. Therefore, the reported FID and CLIP-Score results might be almost identical whether computed in plaintext or ciphertext. Furthermore, in our CipherDM experiments, where performance was evaluated fully under an MPC setting, we still observed clear improvements.
>
> ---
>
> ### **6) Additional cost of fine-tuning compared to vanilla**
>
> Except for the slowdown from using approximated operations instead of vanilla computations, we do not expect additional computational cost.
>
> ---
>
> ### **7) Comparison with fine-tuning**
>
> As we reported in Table 4, inference with CipherDM is also time-consuming, and additional hyperparameter tuning for finetuning is required. We will upload the results as soon as the experiments finish. Aside from this part, if you have any additional questions or follow-up concerns, please feel free to share them, and we will address them.

---

> > ### Comment · Reviewer_tfMV · 2025-11-24
> >
> > I thank the authors for their thorough and clear rebuttal, addressing my key questions.
> >
> > A short follow-up question, in Section 4) of your response to Reviewer f36X, the first column and line of the table states "Loose-Approx(50-step)". Is that supposed to be "Vanilla (50-step)"? The FID and CLIP-Score seems identical to what is reported for "Vanilla" elsewhere.

---

> > > ### Author Response · Authors · 2025-11-24
> > >
> > > Dear Reviewer tfMV,
> > >
> > > We thank you for pointing out the typo. As you noted, the value was mistakenly reported for “Loose-Approx,” and we have corrected it accordingly. Regarding the finetuning results, the experiments are still in progress, and we will share the updated results with you as soon as possible.
> > >
> > > Best Regards,
> > >
> > > Authors

---

> ### Author Response · Authors · 2025-12-01
>
> We would like to share the results and analysis of finetuning experiments.
>
> ---
>
> ### **7) Comparison with fine-tuning**
> To demonstrate that our method outperforms finetuning-based approaches, we conduct finetuning experiments under the CipherDM setting. In this setup, the exponential function in the softmax operation is replaced with CipherDM’s Chebyshev approximation, and then the model is finetuned. Similar to MPCFormer [1], we additionally apply hidden-state distillation at the down, mid, and up blocks.
>
> Finetuning is performed using one-quarter of the original learning rate, and we select the best-performing checkpoint among 500, 1000, 1500, and 2000 iterations. We restrict our search to this range because further training consistently leads to performance degradation. As generating all 10,000 samples is computationally expensive and FID is unreliable with a small sample size, we instead generate 200 samples and evaluate performance using the F1-Score.
>
> As shown in the table below, naive finetuning provides limited improvement, as the approximation gap prevents effective preservation of pretrained knowledge. While distillation mitigates this issue and offers significant gains, **our method still achieves superior performance and remains more efficient, as it requires no additional training**.
>
> | MNIST| F1-Score |
> |---|---|
> | CipherDM | 0.0820 |
> | Finetuning  | 0.0943 |
> | Finetuning + Distillation | 0.4540 |
> | CipherDM + Ours | **0.6000** |
>
> [1] Li, Dacheng, et al. "MPCFORMER: FAST, PERFORMANT AND PRIVATE TRANSFORMER INFERENCE WITH MPC." ICLR, 2023.

---

### Official Review · Reviewer_MYHs · 2025-10-27

**Soundness:** 3
**Presentation:** 3
**Contribution:** 2
**Rating:** 6
**Confidence:** 3

**Summary:**

This paper proposes a novel add-on to an existing privacy-preserving inference method (CipherDM) by addressing the issue of high latency and low fidelity of generated images. The key contribution is that the proposed method leverages the score guidance from the unconditional generation, which provides a higher fidelity signal for image generation. The results show that the image quality has increased after applying the new method on top of CipherDM.

**Strengths:**

1, The paper is well written and easy to follow.

2. The idea of using the unconditional generation as a guidance signal is interesting and novel.

3. Through experiments, the authors have shown that the new add-on to the CipherDM can effectively increase the fidelity of images.

**Weaknesses:**

1, All experiments are conducted on a pretrained Stable Diffusion v1.5 Model, while the CipherDM paper was mainly evaluated on DDPM and DDIM. It might be better to address the transferability issue of the proposed method across different diffusion architectures.

2. In parallel with private inference, another line of research called differentially private diffusion models are serving for the same purpose of privacy preservation. Recent works such as DPDM (https://arxiv.org/abs/2210.09929) and RAPID (https://iclr.cc/virtual/2025/poster/28006) are also trying to provide privacy guarantees to diffusion models. From both papers, I can see that their FID scores on MNIST dataset are much lower than this work or CipherDM. It might be more helpful to address this issue by either comparing their performances through experiments under the same setting or addressing their differences in a separate subsection.

**Questions:**

1. Would it be possible to apply the proposed method in this paper on traditional structures such as DDPM or DDIM? What will be the effect?

2. What is the difference or pros and cons between using private inference or Differentially Private Diffusion Models?

---

> ### Author Response · Authors · 2025-11-20
> **Response to Reviewer MYHs**
>
> We sincerely appreciate your time and effort to review our paper.
>
> ---
>
> ### **1) Performance with traditional architectures**
>
> Our method is already evaluated with two architectures, including Stable Diffusion and CipherDM (which has a similar architecture to DDPM). Since we observe more than 23% reduction in FID on CipherDM, it suggests that our method is likely to work well on traditional architectures such as DDPM as well.
>
> ---
>
> ### **2) Comparison with DP-based approaches**
> We thank the reviewer for pointing out this related line of work. We emphasize that our method and differentially private diffusion models, such as DPDM and RAPID, pursue **different privacy goals** and are therefore complementary rather than directly comparable.
> Specifically, our work targets **privacy-preserving inference**, where a client sends a private input, such as conditions or prompts, to an untrusted model provider, and the goal is to guarantee that the server learns essentially nothing about this input during the inference protocol. In contrast, DPDM and RAPID focus on **protecting the training dataset**. They use DP‑SGD during training so that the final model parameters do not leak information about whether any particular individual’s data was used in training. However, during inference time, the model is used like a standard (non‑private) diffusion model, so user inputs or conditions are visible to the model provider and are not protected by DP.
> We will add these discussions to our revision.

---

### Official Review · Reviewer_f36X · 2025-10-29

**Soundness:** 3
**Presentation:** 3
**Contribution:** 2
**Rating:** 4
**Confidence:** 3

**Summary:**

The paper targets private inference for conditional diffusion under MPC/HE by replacing non-polynomial ops (SiLU, GeLU, exp/softmax) with piecewise polynomial approximations. Relaxing these approximations lowers latency but corrupts the conditional score; the core idea is to use a plaintext, high-fidelity unconditional score (computed without approximations) to correct the encrypted conditional score during sampling. The method derives a guidance update, yielding a simple correction term. Experiments on Stable Diffusion (MS-COCO, Flickr8k) and CipherDM (binary MNIST) show reduced degradation.

**Strengths:**

1. **Clear presentation and reasonable motivation**
    - The paper motivates the problem well: because the text condition is inaccessible during secure inference, the authors refine the conditional path by injecting a clean, uncorrupted unconditional score as a corrective signal.

2. **Simple, architecture-agnostic mechanism**
    - The method is a small, drop-in change to the sampler and is model-agnostic. And it only introduces essentially two scalar knob: the guidance strength and the learning rate.

3. **Good experimental design with clear effectiveness**
     - The experiments probe both sides of the trade-off (latency/communication vs. fidelity) across multiple approximation regimes and datasets, and consistently show that the guidance improves quality to relaxed polynomial approximations.

**Weaknesses:**

1. **Minor quality gains at similar cost**
    - Relative to no score guidance, the improvements are modest, and under the loose approximation, even with the proposed guidance, the generation quality still lags behind the precise approximation without guidance at roughly comparable latency (cf. Tables 1–2).
2. **Incremental contribution; missing comparisons to alternative test-time fixes**
    - The method functions as a test-time refinement/scaling of the sampler, and the “implicit discriminator” view echoes prior discriminator-style interpretations of likelihood models (albeit those are usually applied during training rather than at test time). To justify the novelty and practical value, it would be helpful to compare this approach with other lightweight test-time or post-sampling corrections.

**Questions:**

1. **Iterative rectification within a step**
    - Since the guidance explicitly rectifies the score, can it be applied iteratively within the same time step? If so, does increasing the number of inner refinements (k=1,2,3,…) improve fidelity in a cost-aware manner.

2. **Few-/one-step regimes**
    - As diffusion/flow models move to few-step or even one-step samplers (e.g., consistency models, meanflow), how does the method specialize? In a single-step setting, is there a principled way to form the unconditional reference (score/velocity) and apply your correction so that it still yields a measurable gain?

3. **Budgeted trade-off vs. more steps**
    - The method is a test-time scaling that adds overhead. Under a fixed latency/communication budget, how does guidance compare to simply adding a few extra sampling steps (or slightly tightening the polynomial approximation) without guidance?

---

> ### Author Response · Authors · 2025-11-20
> **Response to Reviewer f36X (1/2)**
>
> We sincerely appreciate the constructive feedback and the opportunity to participate in the discussion.
>
> ---
>
> ### **1) Minor quality gains at similar cost**
>
> While FID reflects distribution-level discrepancies between generated and real datasets, it does not always capture improvements in the quality of individual samples. As shown in Figures 2 and 5, our qualitative study clearly demonstrates that our method produces clean and higher-quality samples compared to the baseline.
>
> Regarding latency, the relative gain is significant (16%) and the absolute reduction is more substantial. Our method reduces the cost of a single forward step by 591 seconds. For a 50-step process, this corresponds to over 8 hours.
>
> ---
>
> ### **2) Comparison with alternative test-time fixes**
>
> To further validate the effectiveness of our approach, we compare it against several lightweight test-time methods, including correction-based approaches [1], degraded-sample–based refinement [2], and test-time scaling strategies [3]. Under the loose approximation setting, **all alternatives perform comparably to or worse than the loose approximation alone** (see table below), since these methods rely on approximated conditional models. In contrast, our method utilizes unconditional generation and therefore avoids these errors. Specifically, PAG [2] requires generating mildly degraded samples, but under approximation errors, the degraded samples become severely corrupted, leading to performance degradation. FKSteering [3], despite using multiple particles, also depends on an approximated verifier network. Thus, verification becomes unreliable and limits its effectiveness.
>
> For latency, a precise measurement requires substantial engineering for all baselines due to the MPC framework's compatibilities. Therefore, we provide a rough comparison. UniPC is expected to have a similar latency to using the loose approximation alone. PAG combined with CFG requires generating additional “bad” samples, leading to 1.5× increase in communication cost. FKSteering requires at least two particles, implying 2× overhead, plus additional verification costs. In contrast, our method introduces only 2.25 seconds per step overhead (0.07% of a single step).
>
> | Flickr8k | FID | CLIP-Score |
> |---|---|---|
> | Vanilla | 21.17 | 0.3369 |
> | Loose-Approx | 23.25 | 0.3347 |
> | Loose-Approx +UniPC(order=3) [1] | 23.84 | 0.3354 |
> | Loose-Approx +PAG(with CFG) [2] | 36.72 | 0.3171 |
> | Loose-Approx +FKSteering(k=2) [3] | 24.42 | 0.3359 |
> | Loose-Approx +Ours | **21.47** | **0.3372** |
>
> [1] Zhao, Wenliang, et al. "Unipc: A unified predictor-corrector framework for fast sampling of diffusion models." NeurIPS, 2023.
>
> [2] Ahn, Donghoon, et al. "Self-rectifying diffusion sampling with perturbed-attention guidance." ECCV, 2024.
>
> [3] Singhal, Raghav, et al. "A General Framework for Inference-time Scaling and Steering of Diffusion Models." ICML, 2025.
>
> ---
>
> ### **3) Iterative rectification within a step**
>
> Our method exploits unconditional generation and utilizes a decaying guidance strength as the sample approaches the signal to keep alignment. Applying the proposed rectification iteratively within the same time step would repeatedly amplify the unconditional component, leading to disrupting this balance and potentially leading to alignment degradation.
>
> ---
>
> ### **4) Few-/one-step regimes**
>
> We additionally evaluate our method in the few-shot regime using the recent consistency model (PCM) [4] at 2, 4, and 8 sampling steps. In the table below, when the number of steps is small, PCM alone with loose approximations suffers from significant performance degradation. However, **when combined with our method, we observe improvements (approximately 9 FID gains at 8 steps and 17 FID gains at 4 steps)**.
> Since our method performs a correction at each step, its effect would become less significant when only a single step (or two steps) is conducted. Nevertheless, under approximation errors, very low-step sampling leads to severe quality drops, making such settings less favorable in practice. From this perspective, our approach offers meaningful benefits for practitioners seeking efficient yet accurate sampling under secure inference.
>
> | Flickr8k | FID | CLIP-Score |
> |---|---|---|
> | Loose-Approx(50-step) | 23.25 | 0.3347 |
> | Loose-Approx +PCM(2-step) | 73.34 | 0.3056 |
> | Loose-Approx +PCM(2-step) +Ours | 73.07 | 0.3059 |
> | Loose-Approx +PCM(4-step) | 53.80 | 0.3183 |
> | Loose-Approx +PCM(4-step) +Ours | 36.69 | 0.3264 |
> | Loose-Approx +PCM(8-step) | 36.69 | 0.3264 |
> | Loose-Approx +PCM(8-step) +Ours | 27.23 | 0.3350 |
>
> [4] Wang, Fu-Yun, et al. "Phased consistency models." NeurIPS, 2024.

---

> ### Author Response · Authors · 2025-11-20
> **Response to Reviewer f36X (2/2)**
>
> ### **5) Budgeted trade-off vs. more steps**
>
> **The per-step overhead introduced by our guidance is extremely small, approximately 0.07% of a single step (2.25 seconds)**. Even over 50 steps, the total additional cost is only about 3.64% (113 seconds), which is negligible compared to the overall latency (a single step requires 3103 seconds). In contrast, adding even one additional sampling step incurs the full cost of a step, resulting in a substantially larger overhead than our method. Similarly, tightening the polynomial approximation requires significant costs. The loose approximation is intentionally designed to minimize expensive operations such as truncation (multiplicative depth). Any relaxation of this approximation would require performing these costly operations more frequently, leading to significantly higher computational overhead than applying our guidance.

---

> > ### Comment · Reviewer_f36X · 2025-11-26
> >
> > I appreciate the authors’ efforts, which have largely addressed my concerns. I am happy to raise my score accordingly.

---

> > > ### Author Response · Authors · 2025-11-29
> > >
> > > Dear Reviewer f36X
> > >
> > > We truly appreciate your constructive feedback and your decision to raise the score to “6: marginally above the acceptance threshold.”
> > >
> > > Best regards,
> > >
> > > Authors

---

### Meta-Review · Area_Chair_2uxM · 2026-01-10

**Summary:**

The paper addresses the significant computational overhead of privacy-preserving inference in diffusion models under Secure Multi-Party Computation. The authors propose a score-correction framework that uses high-fidelity unconditional scores (computed in plaintext) to rectify the errors introduced by loose polynomial approximations in the encrypted conditional path.

Reviewers initially expressed concerns regarding the modest quality gains relative to the cost, the lack of comparison with alternative test-time fixes, and the feasibility of the approach in few-step sampling regimes. Some reviewers also questioned the effectiveness of the method compared to fine-tuning the model with approximated activations. However, the authors' comprehensive rebuttal, which included new experiments on consistency models and fine-tuning benchmarks, addressed the primary technical concerns of the active reviewers.

**Reviewer Concerns:**

Addressed by Rebuttal

- Comparison with Test-time Alternatives: The authors provided a new table comparing their method against UniPC, PAG, and FKSteering. They demonstrated that their method significantly outperforms these alternatives in the "loose approximation" setting because it avoids relying on the corrupted conditional model.
- Few-step Regimes: The authors added experiments using Phased Consistency Models (PCM). They showed meaningful FID improvements (e.g., a 17-point gain at 4 steps) even in low-step settings.
- Fine-tuning vs. Steering: In response to Reviewer tfMV, the authors conducted new experiments showing that their approach outperformed both naive fine-tuning and fine-tuning with distillation while remaining training-free.
- Agnosticism and Scalability: The authors clarified that the method is architecture-agnostic and introduces negligible overhead (approx. 0.07% per step), which addressed concerns about the practical trade-offs of the guidance.

Outstanding Concerns
- While the authors argued that success on CipherDM suggests success on DDPM/DDIM, they did not provide direct comparative experiments for those specific legacy architectures as requested by Reviewer MYHs.

**Reviewer Scores:**

- Reviewer f36X: 4->6
Explicitly states that the authors "largely addressed my concerns. I am happy to raise my score accordingly."
- Reviewer tfMV: 6->8
The reviewer noted they were "more inclined to raising my score than lowering it" and praised the "thorough and clear rebuttal" that addressed their key questions on fine-tuning. Given the superior performance of the steering method over distillation, a move to an 8 is likely.
- Reviewer MYHs: 6->6
The author's clarification on the fundamental difference between private inference and DP-diffusion models likely solidified this reviewer's weak accept stance.

---

### Decision · Program_Chairs · 2026-01-26

Accept (Poster)